# The Genetic Markers of Knee Osteoarthritis in Women from Russia

**DOI:** 10.3390/biomedicines12040782

**Published:** 2024-04-02

**Authors:** Anton Tyurin, Karina Akhiiarova, Ildar Minniakhmetov, Natalia Mokrysheva, Rita Khusainova

**Affiliations:** 1Internal Medicine Department, Bashkir State Medical University, 450008 Ufa, Russia; liciadesu@gmail.com; 2Endocrinology Research Centre, Dmitriya Ulianova Street, 11, 117036 Moscow, Russia; minniakhmetov.ildar@endocrincentr.ru (I.M.); mokrisheva.natalia@endocrincentr.ru (N.M.); ritakh@mail.ru (R.K.); 3Medical Genetics Department, Bashkir State Medical University, 450008 Ufa, Russia

**Keywords:** knee osteoarthritis, candidate genes, locus, GWAS

## Abstract

Osteoarthritis is a chronic progressive joint disease that clinically debuts at the stage of pronounced morphologic changes, which makes treatment difficult. In this regard, an important task is the study of genetic markers of the disease, which have not been definitively established, due to the clinical and ethnic heterogeneity of the studied populations. To find the genetic markers for the development of knee osteoarthritis (OA) in women from the Volga-Ural region of Russia, we conducted research in two stages using different genotyping methods, such as the restriction fragment length polymorphism (RFLP) measurement, TaqMan technology and competitive allele-specific PCR—KASP^TM^. In the first stage, we studied polymorphic variants of candidate genes (*ACAN*, *ADAMTS5*, *CHST11*, *SOX9*, *COL1A1*) for OA development. The association of the *27 allele of the *VNTR* locus of the *ACAN* gene was identified (OR = 1.6). In the second stage, we replicated the GWAS results (*ASTN2*, *ALDH1A2*, *DVWA*, *CHST11*, *GNL3*, *NCOA3*, *FILIP/SENP1*, *MCF2L*, *GLT8D*, *DOT1L*) for knee OA studies. The association of the *T allele of the *rs7639618* locus of the *DVWA* gene was detected (OR = 1.54). Thus, the *VNTR* locus of *ACAN* and the *rs7639618* locus of *DVWA* are risk factors for knee OA in women from the Volga-Ural region of Russia.

## 1. Introduction

Osteoarthritis (OA) is a chronic musculoskeletal condition that primarily affects weight-bearing joints (such as the knees, hips and spine) yet may involve the hands as well as other non-weight-bearing articular sites [1]. More than half of those over age 50 reported knee pain syndrome in the past year [2]. Osteoarthritis is generally a slowly progressive disorder. However, at least one in seven people with incident knee osteoarthritis develops an abrupt progression to advanced-stage radiographic disease, many within 12 months [3]. A large number of studies have been conducted to identify risk factors for the development of osteoarthritis of supporting joints, including the knee joint. It has been found that this pathology is more common in women, and its development is also influenced by age, body weight, trauma to the lower limbs and the knee joint itself, increased physical stress on the joint during work or sports and smoking [4]. In addition to these factors, the development of osteoarthritis of the knee joint is specifically affected by prolonged static posture with bent legs—kneeling and squatting [5].

The present project is a continuation of scientific research, which is being conducted by a team of scientists from the Republic of Bashkortostan, Russia. One of the priority areas of work is the study of clinical and genetic bases of the pathogenesis of diseases of connective tissue and the musculoskeletal system, especially in osteoarthritis. We have previously investigated osteoarthritis both in isolation and in combination with connective tissue dysplasia. We found the association of *rs143383* (*GDF5*) and *rs731236* (*VDR*) with the development of OA and the association of *rs2276455* (*COL2A1*), *rs1544410* and *rs7975232* (*VDR*) with the formation of connective tissue dysplasia in women of Russian ethnicity. Also, we found the association of *rs1544410* and *rs7975232* (*VDR*) with the development of osteoarthritis combined with connective tissue dysplasia. We analyzed 32 polymorphic loci, which, according to databases, are binding sites of various microRNAs to mRNA of target genes and searched for associations with the development of OA and connective tissue dysplasia, considering the localization and age of OA manifestation. We found an association between the T allele of *rs13317* in *FGFR1* and the incidence of the total OA and knee OA in women from the Volga-Ural region of Russia. The G allele of *rs6854081* in *FGF2* was a risk factor for OA development for women of Tatar descent, the C allele of *rs1061237* in *COL1A1* was a risk factor for OA development for Russian women and the T allele of *rs229069* in *ADAMTS5* and the T allele of *rs73611720* in *GDF5* were risk factors for OA development for women of mixed descent [6]. Thus, several ethnospecific genetic markers of osteoarthritis have been identified, which makes it promising to continue research in this direction.

The Volga-Ural region of Russia is characterized by significant ethnic diversity. The indigenous population of the region belongs to various linguistic groups; the Slavic group (Russians) and the Turkic branch of the Altai language family (Tatars) are the most widely represented, followed by the Finno-Ugric peoples in frequency. Thus, the aim of our study was to identify genetic markers of knee osteoarthritis among structural protein genes (*ACAN*, *COL1A1*), catabolic genes (*ADAMTS5*) and genes of enzymes that regulate connective tissue metabolism (*ASTN2*, *ALDH1A2*, *DVWA*, *CHST11*, *GNL3*, *NCOA3*, *FILIP/SENP1*, *MCF2L*, *GLT8D*, *DOT1L*, *SOX9*) in women from the Volga-Ural region of Russia.

## 2. Materials and Methods

### 2.1. *Patient Samples*

We examined 1500 middle-aged (40–60 years) women for the presence of total OA and knee OA. The diagnosis was based on the clinical criteria of the American College of Rheumatology with radiological confirmation. OA was detected in 256 people and knee OA in 134 of them. The control group included 161 women without joint pathology. When forming the sample, patients with primary osteoarthritis of the knee joint were selected. Exclusion criteria were lower extremity trauma and systemic connective tissue diseases. Among the patients included in the study, 17 (12.70%) had undergone total knee replacement. DNA samples from women in the OA and control groups were used as materials for the study. The characteristics of the groups are presented in Table 1. All patients were interviewed regarding the age of onset and duration of disease. The duration of disease ranged from 1 to 38 years (mean 14.93 ± 9.1 years). The ethnic composition of the sample was as follows: 144 (34.53%) were Russian, 159 (38.13%) were Tatar, 30 (7.20%) were Bashkir and 84 (20.14%) were mestizo and representatives of small ethnic groups. Ethnicity was determined on the basis of information about ancestors up to the third generation.

The exclusion criteria were oncological pathology, systemic connective tissue disease, signs of an active inflammatory process of both infectious and non-infectious etiology, a history of traumatic joint injuries, pregnancy (or lactating women) and refusal to participate in the study. The protocol was approved by the Ethics Review Committee of the Bashkir State Medical University, and signed informed consent was obtained from all of the participants.

### 2.2. Genotyping

DNA from peripheral blood was extracted by phenol–chloroform extraction. The DNA concentration and quality of the obtained solution were evaluated using a NanoDrop spectrophotometer (Thermo Scientific, Waltham, MA, USA). The genotyping of the loci was performed using different modifications of the polymerase chain reaction (PCR)–restriction fragment length polymorphism (RFLP) measurement, TaqMan technology and competitive allele-specific PCR—KASP^TM^—a patented technology of LGC Genomics. A detailed description of the primers used, information on fragment lengths and the method for determining single-nucleotide substitutions are described in Table 2.

The genotyping results were obtained using the QuantStudio 12K Flex Real-Time PCR System (Thermo Fisher Scientific, Waltham, MA, USA). To determine the number of repeats of the *VNTR* polymorphic locus of the *ACAN* gene, we performed Sanger sequencing using an Applied Biosystems 3730xl sequencer (Thermo Fisher Scientific, Waltham, MA, USA). The study design as well as the genotyping methods applied to the loci investigated are presented in Figure 1.

The statistical processing of the obtained data was performed using standard packages of Microsoft Excel 2007, Statistica 6.0. Intergroup comparisons of mean values were performed using Student’s *t*-test. The frequencies of alleles and genotypes in patient and control groups were compared by the χ^2^ criterion. For 2 × 2 contingency tables, we used the χ^2^ criterion with Yates’s correction for continuity if the frequency in at least one cell of the table was less than or equal to 5. The degree of association was assessed by odds ratio (OR) values. Correction for multiple comparisons was carried out by calculating the FDR (false discovery rate) using the Benjamini–Hochberg method with the False Discovery Rate Online Calculator (https://tools.carbocation.com/FDR (accessed on 31 March 2024)). The number of pairs of comparisons was taken as the product of the number of markers assessed and the number of comparison groups relative to one control group.

## 3. Results

### 3.1. Study of Candidate Genes

We tested polymorphic loci of aggrecan (*ACAN*), carbohydrate sulfotransferase-11 (*CHST11*), transcription factor-9 (SOX9), aggrecanase-2 (*ADAMTS5*) and collagen type 1 (*COL1A1*). The characteristics of the studied candidate gene loci are shown in Table 3. These candidate genes were chosen on the basis of knowledge about the involvement of their protein products in connective tissue metabolism. The allele and genotype frequencies are shown in Appendix A for *VNTR* and in Appendix A for the rest.

A comparative analysis of allele frequency distributions between patient and control groups revealed statistically significant differences in the frequency of the *27 allele (*VNTR*) between the groups of women with knee OA (0.497) and controls (0.398). The association of allele *27 with knee OA was found (χ^2^ = 4.613; *p* = 0.031; OR = 1.52; 95% CI 1.04–2.23). After adjusting for multiple comparisons, the association did not retain statistical significance. Tendency but not statistically significant differences in the frequency of the 2*27 genotype in the OA group (0.255) and control (0.174) were revealed (χ^2^ = 3.047; *p* = 0.08).

An examination of the *VNTR* polymorphism of the *ACAN* gene revealed 12 allelic variants and 29 genotypes with 19 to 30 repeats, the most frequent alleles with 26 (14.8%), 27 (45.1%) and 28 (32.7%) repeats in the OA group (Appendix A). Alleles with 19, 20, 21, 24 and 30 repeats were rare and occurred in isolated cases. The genotype frequencies of the *VNTR* locus in the group with knee OA and the control group are shown in Figure 2.

To confirm the repeat size, we selected three samples homozygous for the studied locus with a different number of repeats and performed Sanger sequencing. We detected nucleotide variations (ACC-ACT, ACC-GCT, ACT-GCT and ACT-ACT) in 17 and 18 codons within repeats consisting of 57 nucleotides (19 codons). In a study by K.J. Doege and colleagues, they reported polymorphism within repeats; our results are consistent with these data. The ACT-GCT variant is the most common. Thus, there is not only polymorphism in the number of repeats but also a single-nucleotide polymorphism within the locus, resulting in a highly heterogeneous aggrecan protein.

### 3.2. Replication of GWAS Results

We investigated loci selected from GWAS analyses that include genetic variants near genes with an unknown role in OA pathogenesis and chromosomal regions with unknown genes with the highest level of genome-wide significance for replication outcomes. The loci included in the study were as follows: *rs1298744* and *rs2302061* (*DOT1L*), *rs3204689* (*ALDH1A2*), *rs6976* (*GNL3*), *rs11177* (*GLT8D1*), *rs4836732* (*ASTN2*), *rs9350591* (*FILIP1/SENP6*), *rs6094710* (*NCOA3*), *rs11841874* (*MCF2L*), *rs7639618* (*DVWA*), and *rs835487* (*CHST11*) [9,10,11,12,13,14]. The characteristics of the studied loci are presented in Table 4.

The allele and genotype frequencies of the studied loci are presented in Appendix A. A comparative analysis of allele frequency distributions between patient and control groups revealed statistically significant differences in the frequency of the *T allele and *CT genotype of *rs7639618* (*DVWA*) in patients with knee OA compared to controls (0.241 vs. 0.171 and 0.352 vs. 0.230, respectively). The *T allele (χ^2^ = 4.538; *p* = 0.033; OR = 1.54; 95% CI 1.03–2.29) and *CT genotype (χ^2^ = 5.492; *p* = 0.019; OR = 1.82; 95% CI 1.01–3.02) were associated with knee OA. After Benjamini–Hochberg correction, the associations did not retain statistical significance. The protein encoded by this gene binds to β-tubulin [15] and is expressed in joint tissue regardless of the presence of a pathological process, confirming the functional role of the protein product in intracellular chondrocyte transport [16].

## 4. Discussion

### 4.1. History of Osteoarthritis Genetic Marker Research

The study of the genetic basis for osteoarthritis development began with the study of twins and their family members. The first results of such studies appeared in the literature in 1941, when Stecher et al. found an association between the incidence of generalized osteoarthritis of the hands in first-line twins and controls, the incidence of osteoarthritis in monozygotic twins was higher than in dizygotic twins and the genetic component could range from 50 to 65%. However, genetic predisposition has not been fully investigated; twin risk has mainly been calculated. A U.K. study found that the twin risk for the radiologic grading of osteoarthritis on the Kellgren–Lawrence class III scale was 4.99; for osteoarthritis with ankylosis—5.07; and for total arthroplasty patients—8.53.

The next step in the study of the molecular basis of OA was the investigation of candidate genes. A limitation of this type of analysis is that a priori knowledge of the etiology of the disease is required. Another limitation is that only very small regions of the genome can be examined at a time. This means that important genes may be missed when using this method. Using the candidate gene approach, we were able to identify several loci associated with the development of OA in the genes for growth and differentiation factor type 5 (*GDF5)*, estrogen receptor (*ESR*), genes of the transforming growth factor family (*SMAD3*) [17], matrix metalloproteinase-1(MMP1), asporin protein (ASPN), frizzled-related protein (FRZB) and prostaglandin-endoperoxide synthetase (PTGS2) genes and several others [15,18,19,20,21,22,23,24,25,26]. Nevertheless, the use of a candidate gene approach alone has not fully unraveled the mechanisms of genetic predisposition to osteoarthritis.

An important step in the study of genetic predisposition to OA has been the use of the method of searching for genome-wide associations with a high degree of statistical significance (GWAS). GWAS has become an important genetic tool that has allowed researchers to understand the polygenic nature of OA. To date, up to 124 single-nucleotide polymorphisms (SNPs) spanning 95 independent loci have been associated with OA. It should be noted that, as in many other common diseases, most GWAS signaling has been reported in non-coding regions of the genome [27]. Unfortunately, the effect sizes of alleles determining OA risk according to GWAS data are small, most odds ratios (OR) are <1.5 and there are no common loci with a large effect size, such as the human leukocyte antigen (HLA) in autoimmune arthritic diseases such as rheumatoid arthritis. The largest relative risk (RR) score reported so far for the OA locus is 16.70 for a variant in the cartilage oligomeric matrix protein (*COMP*) gene, but this variant is restricted to an extensive Icelandic pedigree and is absent in other European populations [28,29].

Genetic predisposition for knee OA has been deemed relevant, however less so in the hands and hips [30]. Osteoarthritis in general, as well as osteoarthritis of the knee joint, refers to multifactorial diseases, the phenotypic manifestations of which are the result of the interaction of a large number of predictors, including genetic ones. Unfortunately, each of these predictors has an isolated insignificant effect, which does not allow us to identify significant diagnostic markers of disease development. Candidate genes, including *GDF5*, *COL9A1*, *IL1B*, *IL1RN*, *LRCH1*, *CLIP*, *TNA* and *BMP2*, have been reported to be associated with knee osteoarthritis [31]. Some studies indicate the presence of ethnic features in the genetic markers of the development of OA of the knee. A genome-wide significant variant in *LINC01006* (minor allele frequency 12%; *p* = 4.11 × 10^−9^) is less common in European white populations (minor allele frequency < 3%). Five other independent loci reached suggestive significance (*p* < 1 × 10^−6^). These loci are located in or near *MAGI1* (rs145965284, MAF = 27%), *ANKRD6* (rs78571182, MAF = 12%), *EPPK1/PLEC* (rs76983122, MAF = 11%), *PAX7/TAS1R2* (rs4920343, MAF = 13%) and *DDX10/C11orf87* (rs9783397, MAF = 29%); odds ratios ranged from 1.83 to 2.08. Also, no evidence that previously reported OA susceptibility variants in European whites were associated with knee OA in African Americans [32]. Seven case–control studies comprising a total of 3512 knee OA patients and 5405 healthy controls were included in the meta-analysis. There were four studies in an Asian population and three studies in a Caucasian population. A significant association between *rs1871054* (*ADAM12*) and increased knee OA risk was found in each genetic model. No significant association was found between knee OA and *rs3740199*, *rs1044122* or *rs1278279* (*ADAM12*) in any genetic model. There was a modest but statistically significant association between *rs1871054* and knee OA risk in an Asian population, while other polymorphisms in *ADAM12* were not associated with knee OA in any population [33]. The findings suggest the need to investigate genetic markers of knee OA in different populations and ethnic groups, as the ethnospecificity of genetic markers is revealed in musculoskeletal diseases [4].

### 4.2. VNTR

The *VNTR* polymorphism, located in exon 12 of the aggrecan gene, is represented by a varying number of 57 nucleotide tandem repeats encoding 19 amino acids. Alleles ranging from 13 to 34 repeats have been described in the literature. The amino acid sequence encoded by this site contains serine–glicine pairs as two possible attachment points for chondroitin sulfate (CS) chains. The number of alleles may account for up to 30% of the variation in the number of CH chains in a protein molecule [34,35]. The length of the basic protein changes in direct proportion to the number of repeats, and changes in this length can lead to changes in the functional properties of connective tissue and cartilage [36], but available data on the effect of *VNTR* alleles on connective tissue are inconsistent and require further research. The results of studies on the effects of allelic variants of the *VNTR* locus of the *ACAN* gene on cartilage and connective tissue are few and inconsistent. The length of the major protein changes in direct proportion to the number of repeats, and variation in this length can lead to cartilage dysfunction in OA by contributing to the structure of the extracellular matrix and its mechanical properties [35]. Aggrecan molecules with longer chondroitin sulfate domains will have increased density and presumably better osmotic properties. Therefore, one can predict that tissue containing aggrecan with shorter chondroitin sulfate domains may be functionally inferior and more susceptible to mechanical stress. However, this is not always the case. *VNTR* polymorphism in Russian populations has not been studied before. Back pain and degenerative disc lesions were among the first nosologies in which the role of the *VNTR* polymorphism of the *ACAN* gene was studied. The results of these studies are summarized in two meta-analyses. One of them, conducted by J. Gu and colleagues (2013), used data from 965 patients with degenerative intervertebral disc lesions and 982 controls from eight studies. All subjects were divided into three groups based on the number of repeats of the *VNTR* locus—‘short’ alleles (13–25 repeats), ‘normal’ alleles (26–27 repeats) and ‘long’ alleles (28–32 repeats). Short alleles increased the risk of disease by 56% overall (OR 1.54; *p* = 0.03) and up to 65% in patients of the Asian ethnic group (OR = 1.65; *p* = 0.004). An analysis of Caucasians showed no statistically significant associations [37]. In another meta-analysis published a year earlier, G. Xu and colleagues (2012) compared patients with small (less than 23 and 25) and large repeats. Short alleles increased the risk of disease both in the general population and when divided according to ethnicity [38].

Studies about the influence of the *VNTR* locus of the *ACAN* gene on the development of OA began at the end of the 20th century. The work of W.E. Horton and colleagues (1998), in a sample of 93 men (60 years and older), showed that the *27 allele was associated with hand OA (OR = 3.23) but evaluated no statistically significant association between the *27 allele and OA of other localizations, which partially agrees with our association of the *27 allele with knee OA [7]. In contrast, in a study by O.P. Kämäräinen and colleagues (2006) including 630 women in the Finnish population between 45 and 60 years of age, the *27 allele was protective in the development of hand OA. Alleles with more repeats (28–34) in the homozygous state increased the risk of developing the pathology (*p* = 0.036; OR = 1.73), which contradicted our results [39]. P.J. Roughley and colleagues (2006) found no statistical association between the *27 allele and OA in 63 men and women of European descent with hip OA, which also confirms our results [40]. A study of 134 twins by K.M. Kirk and colleagues (2003) showed a non-significant protective effect of the *25 and *26 alleles for the development of knee OA and *28 for the development of hip OA, which is also consistent with our findings [41]. Based on the sequencing results and comparison with the results of K.J. Doege and colleagues (1997), we found a known polymorphism in the 17th and 18th codons of each repeat occurring in all three homozygous samples studied, containing 19 and 26 repeats within each allele. Our studies also confirm the previously described amino acid polymorphism, as a substitution of the first or last nucleotide of the 18th or 19th codon results in Thr/Ser/Ala amino acid alternation [42].

### 4.3. rs7639618 (DVWA)

*DVWA*, on human chromosome 3p24.3, encodes short (276 amino acid) proteins with two regions corresponding to the von Willebrand factor type A (VWA) domain, which was presented in a variety of proteins and has a role in cell adhesion, membrane transport and protein–protein interactions. DVWA interacts with β-tubulin and such interaction is thought to be a protective factor in OA pathogenesis. The strength of the binding is reduced by rs7639618 in the VWA domain. Such weaker binding between β-tubulin and the wild protein may be at a higher risk of developing OA [1]. Studies of *rs7639618’*s influence on OA development give different information, probably due to significant differences in allele frequencies among different ethnic groups. The highest of the minor allele frequency (*T, in some sources—*G) reaches 50% in the populations of Southeast Asia (https://www.ensembl.org, accessed on 31 March 2024). In a study by Miyamoto et al. (2008), the minor allele *G of *rs7639618* showed an association with knee OA in Japanese cohorts, as well as replication studies of the Chinese Han cohort (*p* = 7.3 × 10^−11^, OR = 1.43; 95% CI 1.28–1.59) [11], which contradicts our results. The results of one of the meta-analyses indicated a significant association between *rs7639618* polymorphism and OA susceptibility observed in dominant and co-dominant models. In the subgroup analysis, we found that *rs7639618* and rs11718863 polymorphisms of the *DVWA* gene were associated with OA risk in Asians (GG + GA vs. AA: OR = 1.34; *p* < 0.001; G vs. A: OR = 1.29; *p* = 0.019). Furthermore, for *rs7639618*, the dominant (GG + GA) and heterozygote (GA) variants may strongly increase knee OA susceptibility (GG + GA vs. AA: OR = 1.27; *p* < 0.001) in Asians [43]. In a meta-analysis that included 9500 OA cases and 9365 controls in seven case–control studies, Wang et al. observed a significantly increased risk of knee OA susceptibility in an allelic comparison in Asians (A versus G: OR = 1.16; 95% CI 1.04–1.30), while in Caucasians, even with an increased sample size, there was no significant association in any allelic models [44]. In the combined population, the differences did not reach the level of statistical significance, having the character of a trend (*p* = 0.06); nevertheless, this may indicate a potential association of the *rs7639618* locus with knee osteoarthritis. After dividing the population by ethnicity into patients of European and Asian origin, a statistically significant association was obtained for the Asian cohort (*p* = 0.02), while the significance level decreased for the European cohort (*p* = 0.66). Thus, the association of the *rs7639618* polymorphic variant was found only for patients of Asian origin, which is consistent with a number of previous findings [45,46]. Such differences are due to the high heterogeneity of OA, including ethnicity; often, the same genetic markers can have different effects in different populations **[47]**. The frequency of the G allele of the *rs7639618* polymorphic variant was higher in the control group of the European cohort compared to that in the Asian cohort (84.3% and 53.7%, respectively) [48]. However, in a study of 61 Sicilian patients with knee OA and 100 healthy subjects, it is possible to note that the *rs7639618* genotype H  +  Mut (*CT + *TT) are more represented in the group with a more severe OA radiographic grade (55%) [49].

## 5. Conclusions

The current study demonstrated associations between the *27 allele of the *VNTR* polymorphism in *ACAN* and the *T allele of *rs7639618* in *DVWA* and the incidence of knee OA in women from the Volga-Ural region of Russia.

## Figures and Tables

**Figure 1 biomedicines-12-00782-f001:**
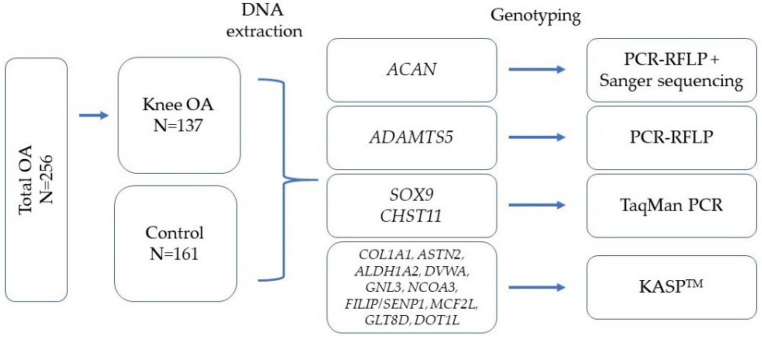
Study design and genotyping methods for loci analyzed. Statistical processing and software.

**Figure 2 biomedicines-12-00782-f002:**
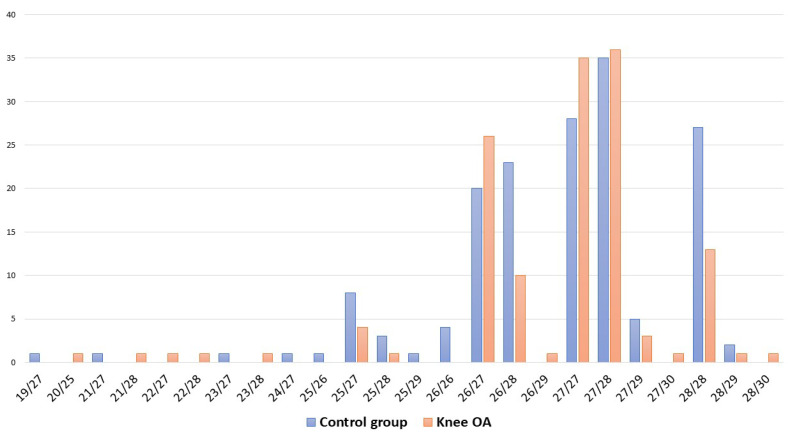
Frequencies of VNTR genotypes of *ACAN* gene polymorphism in women with knee OA and controls.

**Table 1 biomedicines-12-00782-t001:** Age of study subjects.

	*n*	Age, Years		
		M ± m	Min	Max
Total OA	256	55.74 ± 8.64 *p* = 0.51	20	86
Knee OA	137	57.13 ± 7.98 *p* = 0.14	26	86
Control group	161	45.55 ± 12.55	32	73

**Table 2 biomedicines-12-00782-t002:** Primer sequences and detection methods for the studied loci.

Gene	Locus	Primer Sequence (5′-3′), Fragment Length, Annealing t^0^.	Detection Method	Link
Candidate genes				
*ACAN*	*VNTR*	*5′-TAGAGGGCTCTGCCTCTGGAGTTG-3′* *5′-AGGTCCCCTACCGCAGAGGTAGAA-3′* t^0^: 58 °C PCR product length: 1143–1770 n.p.	6% polyacrylamide gel	Doege K.J., 1997 [7]
*ADAMTS5*	*rs226794*	*5′-TCCTCCACATACTCCGCACT-3′* *5′-CAAAATCTGCTTTCTGGCAAT-3′* t^0^: 59 °C fragment length—330 n.p. **GG—*105 и 225 n.p. **AG—*330, 105 и 225 n.p. **AA—*330 n.p.	PCR/RFLP MbiI	Jiaao Gu., 2013 [8]
	*rs2830585*	*5′-GCCTGGACAACAGTGTGAGA-3′* *5′-GGAGTGCAGTTTGCCTATCG-3′* t^0^: 65 °C fragment length—310 n.p. **TT*—310 n.p. **CT*—310, 241, 69 n.p. **CC*—241, 69 n.p.	PCR/RFLP BsuRI	Jiaao Gu., 2013
*SOX9*	*rs1042667*	Primers: *FJ:CCAGAACTCCAGCTCCTA* *RJ:CTGGTTGGTCCTCTCTTTC* Probes: FAM-aagggCgaAgaTggc-BHQ-1 VIC-aagggCgaCgaTggc-BHQ-2 t^0^: 62 °C	TaqMan PCR	Own development
	*rs2229989*	Primers: *FJ:GAACGAGAGCGAGAAGCG* *RJ:GGAGATGTGCGTCTGCTC* Probes: FAM-agaaGgaCcaTccgga-BHQ-1 VIC-agaaGgaCcaCccgga-BHQ-2 t^0^: 64 °C	TaqMan PCR	Own development
	*rs7217932*	*FJ: AAGGCTTATTATATGTTAGAA* *RJ: GTCCAAGTTGATTTTTTC* FAM-cagCactTcttGtaga-BHQ-1 VIC-cagCactCcttGtaga-BHQ-2 t^0^: 56 °C	TaqMan PCR	Own development
*CHST11*	*rs6539153*	Primers: *FJ: CCAACTCCATGATCTCTG* *RJ: TGACCTCTCACCTCATAG* Probes: FAM-tcagaaGtcTaaTtccctgt-BHQ-1 VIC-tcagaaGtcCaaTtccctgt-BHQ-2 t^0^: 60 °C	TaqMan PCR	Own development
*COL1A1*	*rs1107946*	*GTCAGTTCCAAGAGA[A/C]CCCCTCCCTAATAGG*	KASP^TM^	Own development
	*rs1800012*	*CCTGCCCAGGGAATG[G/T]GGGCGGGATGAGGGC*	KASP^TM^	Own development
Replication of GWAS results				
*ASTN2*	*rs4836732*	*GAGAGACAGCACCTA[C/T]TTTCTGAGGTCTAAG*	KASP^TM^	Own development
*DVWA*	*rs7639618*	*CATTGACCCCTACCA[C/T]ATAACACAACAACTC*	KASP^TM^	Own development
*MCF2L*	*rs11842874*	*TAATGTATGGTGACA[A/G]GAGTCGGGATGGGGC*	KASP^TM^	Own development
*GLT8D1*	*rs6976*	*AACTGTTACTTCCCA[C/T]GCATGCTATCTTCCA*	KASP^TM^	Own development
*DOT1L*	*rs2302061*	*GCCCGGGACCGCGAG[C/G]TCGACCTCAAGAATG*	KASP^TM^	Own development
	*rs12982744*	*TCGGCTGTGGGCACC[C/G]GACATGTGGCTGGCG*	KASP^TM^	Own development
*ALDH1A*	*rs3204689*	*GGAGCTGGTACACTA[C/G]AGATGTAGTAAGAAC*	KASP^TM^	Own development
*GNL3*	*rs11177*	*GGCTTCTTGTGACCC[C/T]GCTTTTTAGCCTCCT*	KASP^TM^	Own development
*FILIP1/* *SENP6*	*rs9350591*	*GTTGACAACATGAAC[C/T]GGAGACAAGAAATAA*	KASP^TM^	Own development
*NCOA3*	*rs6094710*	*GGGCTGTCTGCACGT[A/G]CAATGTGTTTATTGG*	KASP^TM^	Own development
*CHST11*	*rs835487*	Primers: *FJ, GACTCTGTCTGCATCACA* *RJ, GGTCAACTGGAATGTTCTG* Probes: FAM-cgttttAaaggTacctCctatt-BHQ-1 VIC-cgttttAaaggCacctCctatt-BHQ-2 t^0^: 60 °C	TaqMan PCR	Own development

Note: * t^0^ primer annealing for loci genotyped using KASP^TM^ technology: 61–55 °C, n.p.—nucleotide pairs.

**Table 3 biomedicines-12-00782-t003:** Characterization of studied loci in candidate genes.

Locus	Gene	Chromosomal Localization	H_pred_	H_obs_	HW_pval_	MAF
*VNTR*	*ACAN*	15q26.1	0.665	0.629	0.998	0.001
*rs6539153*	*CHST11*	12q23.3	0.481	0.476	0.935	0.402
*rs226794*	*ADAMTS5*	21q21.3	0.261	0.284	0.183	0.155
*rs2830585*	*ADAMTS5*	21q21.3	0.199	0.192	0.706	0.112
*rs1042667*	*SOX9*	17q24.3	0.477	0.456	0.483	0.392
*rs2229989*	*SOX9*	17q24.3	0.363	0.363	1	0.238
*rs7217932*	*SOX9*	17q24.3	0.499	0.478	0.494	0.483
*rs1107946*	*COL1A1*	17q21.33	0.324	0.306	0.390	0.203
*rs1800012*	*COL1A1*	17q21.33	0.253	0.240	0.465	0.148

Note: hereafter, H_obs_—observed heterozygosity, H_pred_—expected heterozygosity, HW_pval_—*p*-value for assessing compliance with Hardy–Weinberg equilibrium (maintained at *p* > 0.05) and MAF—frequency of minor allele in sample we studied.

**Table 4 biomedicines-12-00782-t004:** Characteristics of the GWAS loci.

Locus	Gene	Chromosomal Localization	H_pred_	H_obs_	HW_pval_	MAF	GWAS *p*-Value	GWAS OR
*rs6976*	*GLT8D1*	3p21.1	0.494	0.484	0.742	0.446	1.2 × 10^−10^	1.12 (1.08–1.16
*rs11177*	*GNL3*	3p21.1	0.495	0.489	0.880	0.448	7.2 × 10^−11^	1.12 (1.08–1.16)
*rs7639618*	*DVWA*	3p25.1	0.327	0.320	0.731	0.206	7.3 × 10^−8^	1.54 (1.32–1.81)
*rs9350591*	*FILIP/SENP6*	6q14.1	0.209	0.218	0.554	0.119	2.42 × 10^−9^	1.18 (1.12–1.25)
*rs4836732*	*ASTN2*	9q33.1	0.500	0.559	0.022	0.500	6.1 × 10^−10^	1.20 (1.13–1.27)
*rs835487*	*CHST11*	12q23.3	0.416	0.425	0.800	0.295	1.64 × 10^−8^	1.13 (1.09–1.18)
*rs11842874*	*MCF2L*	13q34	0.227	0.223	0.821	0.131	2 × 10^−8^	1.22 (1.14–1.30)
*rs3204689*	*ALDH1A2*	15q21.3	0.471	0.480	0.796	0.379	8.6 × 10^−11^	1.46 (1.31–1.63)
*rs1298744*	*DOT1L*	19p13.3	0.454	0.492	0.121	0.349	1.1 × 10^−11^	1.17 (1.11–1.23)
*rs2302061*	*DOT1L*	19p13.3	0.250	0.254	0.913	0.146	1.1 × 10^−11^	1.18 (1.13–1.35)
*rs6094710*	*NCOA3*	20q13.12	0.104	0.101	0.714	0.055	7.9 × 10^−9^	1.20 (1.08–1.34)

## Data Availability

Data are contained within the article and Appendix A.

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
