# Peer review of "The Genetic Markers of Knee Osteoarthritis in Women from Russia"

_biomedicines, 2024, doi:10.3390/biomedicines12040782_

Round 1

Reviewer 1 Report

Comments and Suggestions for Authors

This study by Tyurin et al investigated the genetic markers for the development of knee OA in women from the Volga-Ural region of Russia. They revealed ACAN, ADAMTS5, CHST11, SOX9, COL1A1 as important factor for the OA development in the first stage. Association of the *27 allele of the VNTR locus of the ACAN gene was identified (OR=1,6). At the second stage, they replicated the GWAS results (ASTN2, ALDH1A2, DVWA, CHST11, GNL3, NCOA3, FILIP/SENP1, MCF2L, GLT8D, DOT1L) for knee OA studies. Association of the *T allele of the rs7639618 locus of the DVWA gene was detected. However, authors need to revise some major points, as listed below, to make this study complete.

1.     I suggested authors could provide a step-by-step workflow as a figure in methods section to make us realize the result.

2.     I suggested authors could provide more details about the recruited patients’ such as knee surgery history, knee injury history etc.

3.     Authors should provide OR value in the table 3.

4.     Is their any possible molecular mechanism literature that could support your finding?

I suggest authors could replenish in the discussion section.

Comments on the Quality of English Language

Minor editing of English language required

Reviewer 2 Report

Comments and Suggestions for Authors

1. The abstract is extremely short and lacks essential details (background, aim, methods, results, conclusion).

2. The introduction should present a background of the addressed topic rather than listing the results of previous studies.

3. The aim should be written in more details at the end of the introduction.

4. Table 1 should be renamed as "age of the study subjects" rather than "characteristics". Also, the age should be statistically compared between the three study groups.

5. The methods are very short and lack essential details. All the supplementary data should be included in the manuscript (not as supplementary).

6. Lines 122-131 should be included in the introduction rather than the results.

7. Lines 144, 166-168 should be included in the discussion.

Comments on the Quality of English Language

English is accepted. Minor editing may be required.

Reviewer 3 Report

Comments and Suggestions for Authors

Yet another association analysis with a low sample size replicating studies already published. We need to start finding ways to analyse the data that we already have instead of just publishing endless and meaningless association studies. 

Comments on the Quality of English Language

I didn't detect any excessively bad english.

Round 2

Reviewer 1 Report

Comments and Suggestions for Authors

The current form is ok

Author Response

We thank the honorable reviewer for the positive evaluation of the changes made to the manuscript

Reviewer 2 Report

Comments and Suggestions for Authors

The authors tried to reply to my comments, however, some of the modifications were not appropriate.

1. I asked for a background in the introduction, however, what the authors have done is adding a very prolonged list of results of previous irrelevant studies.

2. I asked for a detailed aim, instead, the authors added a long irrelevant sentence "The Volga-Ural region of Russia is characterized by significant ethnic diversity. The indigenous population of the region belongs to various linguistic groups, the Slavic group (Russians) and the Turkic branch of the Altai language family (Tatars) are the most widely represented, followed by the Finno-Ugric peoples in frequency", with no further clarification of the aim.

Comments on the Quality of English Language

Minor revision is required.
